# Integrated Framework for Detecting the Areas Prone to Flooding Generated by Flash-Floods in Small River Catchments



**Romulus Costache** [1,2,3,†] [ID], **Alina Barbulescu** [3,†] [ID] **and Quoc Bao Pham** [4,*]

1  Research Institute of the University of Bucharest, 90-92 Sos, Panduri, 5th District, 050663 Bucharest, Romania; romuluscostache2000@yahoo.com

2  National Institute of Hydrology and Water Management, București-Ploiești Road, 97E, 1st District, 013686 Bucharest, Romania

3  Department of Civil Engineering, Transilvania University of Brasov, 5, Turnului Str, 500152 Brasov, Romania; alinadumitriu@yahoo.com

4  Institute of Applied Technology, Thu Dau Mot University, Thu Dau Mot City 75000, Binh Duong Province, Vietnam

*  Correspondence: phambaoquoc@tdmu.edu.vn

†  These authors contributed equally to this work.

**Abstract:** In the present study, the susceptibility to flash-floods and flooding was studied across the Izvorul Dorului River basin in Romania. In the first phase, three ensemble models were used to determine the susceptibility to flash-floods. These models were generated by a combination of three statistical bivariate methods, namely frequency ratio (FR), weights of evidence (WOE), and statistical index (SI), with fuzzy analytical hierarchy process (FAHP). The result obtained from the application of the FAHP-WOE model had the best performance highlighted by an Area Under Curve—Receiver Operating Characteristics Curve (AUC-ROC) value of 0.837 for the training sample and another of 0.79 for the validation sample. Furthermore, the results offered by FAHP-WOE were weighted on the river network level using the flow accumulation method, through which the valleys with a medium, high, and very high torrential susceptibility were identified. Based on these valleys' locations, the susceptibility to floods was estimated. Thus, in the first stage, a buffer zone of 200 m was delimited around the identified valleys along which the floods could occur. Once the buffer zone was established, ten flood conditioning factors were used to determine the flood susceptibility through the analytical hierarchy process model. Approximately 25% of the total delimited area had a high and very high flood susceptibility.

**Keywords:** flooding; flash-floods; bivariate statistics; fuzzy multicriteria decision-making; small catchments; Romania

## 1. Introduction

According to Hu et al. [1], a total number of 2.5 billion peoples were affected by flash-floods and floods between 1994 and 2013. In the same period, 0.16 billion fatalities occurred due to the same natural risk phenomena. Therefore, since flash-floods are extremely severe phenomena, they are also very dangerous for human life [2,3]. These phenomena appear most frequently in small river basins characterized by a high slope. Additionally, areas with smaller slopes favor the accumulation of the transported water and materials [4]. In this context, the identification of sections that favor the surface runoff occurrence, torrential valleys on which the flash-floods are propagated, and the flood susceptibility assessment in those regions, is one of the most important measures to combat the negative effects of these phenomena on water quality and human society. Additionally, the results provided by this type of analysis are very important in assessing a region's vulnerability and risk to flash floods. It should be noted that most of the procedures regarding the evaluation of flash-flood and flood risk assessment, which were adopted by the European countries includes

the use of several traditional methods such as hydraulic and hydrological modeling. These techniques are time consuming and very expensive [5,6]. In this context, the need to find faster, more accurate, and cheaper techniques for determining the flood hazard has significantly increased.

In recent years, the scientific field of flash-food and flood susceptibility assessment has had a high dynamic due to the fast development of the techniques and software used to perform these analyses [7]. Thus, to assess the flood susceptibility, Geographic Information System (GIS) techniques, complex models of bivariate statistics, and machine learning are used [8]. The most used bivariate statistical techniques for assessing susceptibility to natural hazards are weights of evidence [9], frequency ratio [10], evidential belief function [11], certainty factor [12], statistical index [13], and index of entropy [14]. The most well-known machine learning models used in the study of susceptibility to floods are decision trees [15], multilayer perceptron [16], logistic regression [17], support vector machine [18], bagging [19], k-nearest neighbor [20], naïve Bayes [21], Decorate [22], Dagging [15], and adaptive neuro-fuzzy inference system [23]. Many researchers have assessed the risk of flash-floods and floods by using ensemble models resulting from the combination of several methods [15,21,24].

Nevertheless, in all the research papers where machine learning and bivariate statistics were used, the susceptibility was estimated separately for flash-floods and flooding. Up to now, there is no approach in which the susceptibility to these two phenomena, which are strongly related, can be estimated together. A first attempt to identify the torrential valleys, based on the flash-flood susceptibility, was done by Costache et al. [25]. In that study, the authors managed to detect the river valleys prone to flash-flood propagation using four hybrid models and the flow accumulation method. Nevertheless, the flooding susceptibility was not estimated along the torrential river valleys, this fact being a shortcoming that should be addressed.

In this context, we aimed to propose an integrated approach for estimating the surface runoff susceptibility and the susceptibility to floods. Thus, in the first stage, we follow the identification of areas susceptible to flash-floods by applying three overall models generated by combining frequency ratio, statistical index, and weights of evidence bivariate statistics models, on the one hand, and fuzzy analytical hierarchy process on the other hand. The models' performances were evaluated utilizing the ROC curve. The second stage of the study aims to identify the torrential valleys susceptible to the propagation of the upstream flash-floods by applying the flow accumulation method. Once the valleys with a medium, high, and very high potential for flash-flood propagation are identified, the flood susceptibility is calculated to determine the areas exposed to floods. Flood susceptibility is determined through the analytical hierarchy process stand-alone model.

It should be mentioned that this is the first time in the literature when the susceptibility of these two phenomena, flash-floods and flooding generated by them, were analyzed in an integrated way and in a spatial causal relationship. The previous studies carried out in Romania as well as in any part of the globe were focused on the estimation of flooding or flash-flood susceptibility without taking into account their strong spatial relationship.

## 2. Study Area

The present study focused on the Izvorul Dorului River basin located in the mountainous area of the central part of Romania. The surface of the study area is 33 km$^2$, which falls into the category of small-area basins that are frequently affected by rapid floods. The altitude inside the study zone varies from 763 m to 2202 m (Figure 1a). This high difference in altitude on a small area creates favorable premises for flash-flood genesis and their propagation from the upper to the lower part of the river basin. The river basin is characterized by an average high slope of 15.6°, which is another indicator of the high potential for flash-flood propagation along the valleys in the study area. According to the existing information and, as can be seen in Figure 1b, the afforestation degree of the river basin is around 50%. Additionally, from Figure 1b, one can remark that in the perimeter of

the deforested surfaces located at the highest altitude also exists a very high potential for a rapid surface runoff on the slopes. This is another element indicating that the genesis of the flash-floods is related to the high-altitude region of the river basin from where they are propagated along the steep river valleys toward the lowland area. The lower part of the study area corresponds to the built space of Sinaia city, the most famous mountain tourist resort in Romania. This locality has been affected by floods multiple times, caused by Izvorul Dorului River and its tributaries. One of the most violent flash-floods took place in August 2010 when several dozens of buildings were affected as well as National Road 1, National Road 71, and the railroad between Bucharest and Brasov cities. Additionally, as a result of different strong floods, several landslides were activated and affected the houses from Sinaia.

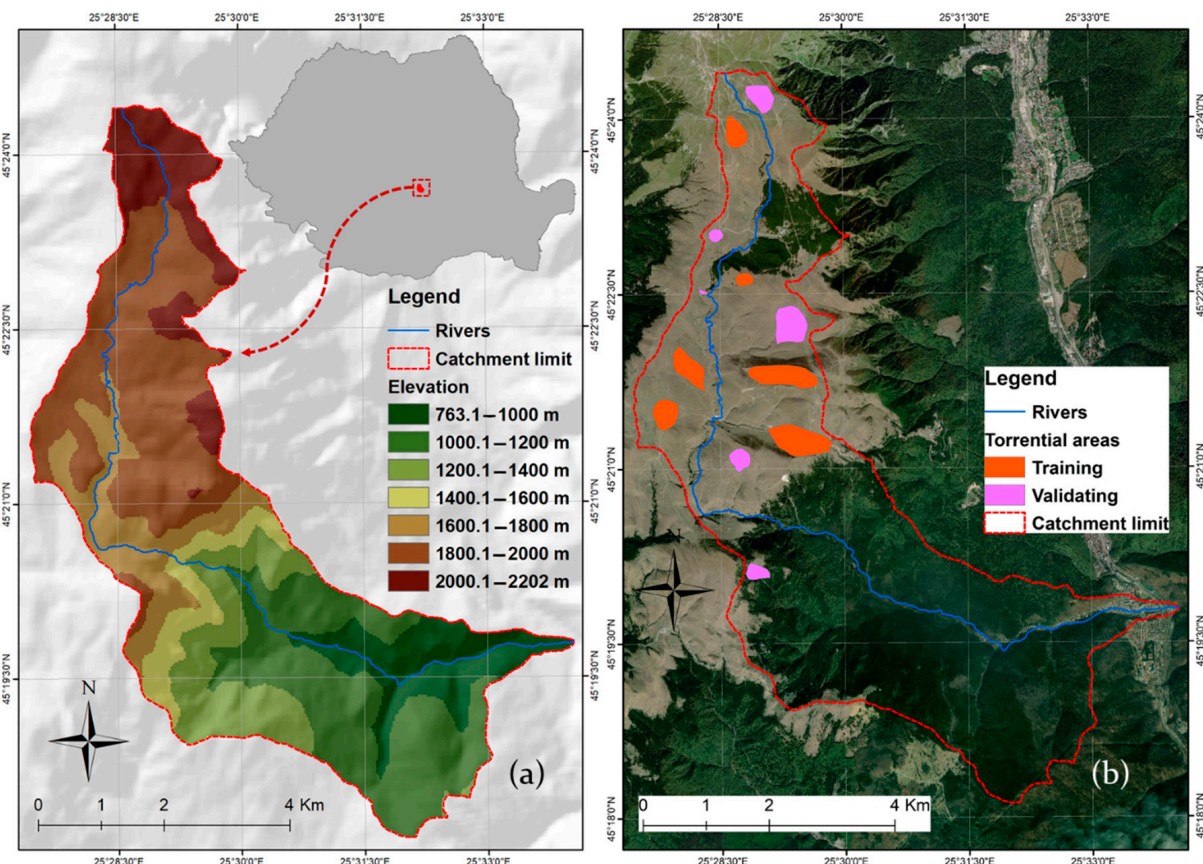

**Figure 1.** The study area. (**a**) Study area location in Romania; (**b**) torrential area location.

## 3. Methods

### 3.1. Background of the Models

#### 3.1.1. Statistical Index

Proposed by van Westen [26], statistical index (SI) is a bivariate method widely used in natural risk susceptibility evaluation studies [13,27]. According to this model, the score of a predictor class can be computed by applying the natural algorithm to the ratio between the density of pixels associated with the phenomenon presence in the predictor class and the density of the same pixels across the study area [28]. Thus, a well-known formula to estimate the SI weight is the following:

$$W_{ij} = \ln\left(\frac{f_{ij}}{f}\right) = \ln\left(\frac{\frac{Npix(S_i)}{Npix(N_i)}}{\frac{\sum Npix(S_i)}{\sum Npix(N_i)}}\right), \tag{1}$$

where $W_{ij}$ is the weight of the class/category $i$ of predictor $j$; $f_{ij}$ is the density of the phenomenon in class $i$ of predictor $j$; $f$ is the density of phenomenon in the study; $Npix(Si)$ is the number of pixels associated with the phenomenon in class $i$; and $Npix(Ni)$ is the sum of pixels of the same parameter class.

### 3.1.2. Frequency Ratio

Frequency ratio ($FR$) is a bivariate statistical model, widely applied to evaluate flood and landslide susceptibility mapping worldwide [9,10,13]. The relationship between food occurrences and conditioning parameters is used to analyze and calculate the frequency ratio. The mathematical expression of frequency ratio (i.e., the frequency ratio of class $i$ of factor $j$) is given in Equation (2) [10]:

$$FR = \frac{\frac{Npix(1)}{Npix(2)}}{\frac{\sum Npix(3)}{\sum Npix(4)}} \tag{2}$$

where $Npix(1)$ is the total number of torrential points contained by a class/category of factor; $Npix(2)$ is the total number of pixels contained by each class/category; $\sum Npix(3)$ is the total number of torrential pixels within the study area; and $\sum Npix(4)$ is the total number of pixels within the study area.

After calculating the frequency ratio, each controlling factor summed up all the values to generate a map of flood vulnerability. If the frequency ratio is greater than 1, the conditioning factors strongly influence flooding, otherwise, there is a negative relationship between conditioning factors and flood occurrence.

### 3.1.3. Weights of Evidence

Weights of evidence (WOE) is a widely used statistical model for landslide, flood, and fire forest susceptibility assessment [29–31]. This method was first introduced for geological studies in 1992, then adopted for the analysis of different hazards (e.g., fire forest, flood, landslides) [27]. This method estimates the weights of evidence coefficients based on the relationship between each class of factors and the flood absence/presence. The positive weight ($W^+$) and the negative weight ($W^-$) are necessary for the computation. These weights reflect the presence and absence of areas affected by the flood, respectively, and can be computed using the following [29–31]:

$$W^+ = ln\frac{P\{B|S\}}{P\{B|\overline{S}\}} \tag{3}$$

$$W^- = ln\frac{P\{\overline{B}|S\}}{P\{\overline{B}|\overline{S}\}} \tag{4}$$

where $B$ and $\overline{B}$ are the presence and absence of flood conditioning parameters, respectively; $P$ is the probability; and $S$, and $\overline{S}$ are the presence and absence of flooding, respectively.

The output of the performed processes is used to implement Equations (3) and (4) in ArcGIS. Subsequently, the mathematical representation of these two equations are [29]:

$$W^+ = ln\frac{\frac{Npix_1}{Npix_1 + Npix_2}}{\frac{Npix_3}{Npix_3 + Npix_4}} \tag{5}$$

$$W^- = ln\frac{\frac{Npix_2}{Npix_1 + Npix_2}}{\frac{Npix_4}{Npix_3 + Npix_4}} \tag{6}$$

where $W^+$ and $W^-$ are the positive and negative weights, respectively; $Npix_1$ and $Npix_2$ are the number of pixels with flood points inside *and* outside of the class, respectively;

and $Npix_3$ and $Npix_4$ are the number of pixels without flooding inside and outside of the class, respectively.

The final weights of evidence coefficients ($Wf$) assigned to each factor class can be obtained as follows [29]:

$$Wf = Wplus + Wmintotal - Wmin \tag{7}$$

where ($Wf$) is the final weight of evidence coefficients; $Wmintotal$ is the total of all negative weights in a multiclass map; and $Wplus$ and $Wmin$ are the positive negative weights of a class factor, respectively.

### 3.1.4. Fuzzy Analytical Hierarchy Process

The analytical hierarchy process (AHP) is an algorithm used for flood, landslide, and fire forest susceptibility mapping [32–35]. Through a pairwise comparison matrix constructed based on expert knowledge, AHP was used to calculate the weights of relevant criterion map layers. Since AHP has several advantages such as its fuzzy extension, the fuzzy analytical hierarchy process (FAHP) was proposed and applied to solve the hierarchical fuzzy problems. It can be employed to increase the analysis quality, reducing the subjectivity in the estimation of weights criteria by a combination of the fuzzy set theory and the analytical hierarchy process [36]. The following steps show how to determine the weights of criteria in the FAHP.

The pairwise comparison matrices are constructed from flood conditioning factors (elevation, slope angle, stream density, curve number, rainfall, lithology, land use, soil texture, etc.). Linguistic terms are assigned to the pairwise comparison (Equation (8)) to establish the most important criteria [37]:

$$A' = \begin{bmatrix} 1' & a'_{12} & \cdots & a'_{1n} \\ a'_{21} & 1' & \cdots & a'_{1n} \\ \vdots & \vdots & \ddots & \vdots \\ a'_{n1} & a'_{n2} & \cdots & 1' \end{bmatrix} = \begin{bmatrix} 1' & a'_{12} & \cdots & a'_{1n} \\ 1/a'_{21} & 1' & \cdots & a'_{1n} \\ \vdots & \vdots & \ddots & \vdots \\ 1/a'_{n1} & 1/a'_{n2} & \cdots & 1' \end{bmatrix} \tag{8}$$

where $a'_{ij}$ indicates a pair of criteria $i$ and $j$.

The Buckley method [38] is utilized to calculate the fuzzy geometric mean and fuzzy weight of each criterion by:

$$r'_i = \left( a'_{i1} \otimes a'_{i2} \otimes \ldots \otimes a'_{in} \right)^{\frac{1}{n}}, \tag{9}$$

$$w'_i = r'_i \otimes \left( r'_1 \otimes \ldots \otimes r'_n \right)^{-1}, \tag{10}$$

where $a'_{in}$ is the fuzzy comparison value between the pair criterion $i$ and criterion $n$; and $r'_1$ is the geometric mean of the fuzzy comparison values for criterion $i$ compared to each of the other criteria; $w'_i$ is the fuzzy weighting of the $i$th criterion; and $w'_i = (lw_i, mw_i, uw_i)$, where $lw_i$, $mw_i$ and $uw_i$ are the values of the lower, middle, and upper, fuzzy weighting of the $i$th criterion, respectively [37,39].

The extent analysis algorithm was applied to determine the final values of the flood conditioning factor weights. The construction of a fuzzy triangular comparison matrix is the first step. This matrix is done by [40]:

$$A' = \left( a'_{ij} \right)_{nxn} = \begin{bmatrix} (1,1,1) & (l_{12}, m_{12}, u_{12}) & \cdots & (l_{1n}, m_{1n}, u_{1n}) \\ (l_{21}, m_{21}, u_{21}) & (1,1,1) & \cdots & (l_{2n}, m_{2n}, u_{2n}) \\ \vdots & \vdots & \ddots & \vdots \\ (l_{n1}, m_{n1}, u_{n1}) & (l_{n2}, m_{n2}, u_{n2}) & \cdots & (1,1,1) \end{bmatrix} \tag{11}$$

where $a'_{ij} = (l_{ij}, m_{ij}, u_{ij})$ and $a'^{-1}_{ij} = (1/l_{ij}, 1/m_{ij}, 1/u_{ij})$ for $i, j = 1, \ldots, n$ and $i \neq j$.

Next, we computed the priority vector of the triangular matrix. Then, the fuzzy arithmetic function was employed to sum up each row of the matrix $A'$ in a first stage, as follows:

$$RS_i = \sum_{j=1}^{n} a'_{ij} = \left( \sum_{j=1}^{n} l_{ij}, \sum_{j=1}^{n} m_{ij}, \sum_{j=1}^{n} u_{ij} \right), \ i = 1, \ldots, n \quad (12)$$

Then, the value of the fuzzy synthetic extent in terms of the $i$th object is obtained through the normalization of the above relation, as follows [32]:

$$S'_i = \sum_{j}^{n} a'_{ij} \otimes \left[ \sum_{k=1}^{n} \sum_{j=1}^{n} a'_{kj} \right]^{-1} = \left( \frac{\sum_{j=1}^{n} l_{ij}}{\sum_{k=1}^{n} \sum_{j=1}^{n} u_{kj}}, \frac{\sum_{j=1}^{n} m_{ij}}{\sum_{k=1}^{n} \sum_{j=1}^{n} m_{kj}}, \frac{\sum_{j=1}^{n} u_{ij}}{\sum_{k=1}^{n} \sum_{j=1}^{n} l_{kj}} \right), \ i = 1, \ldots, n. \quad (13)$$

The computation of the degree of possibility of $S'_i \geq S'_j$ represents the third step and is achieved through Equation (14):

$$V(S'_i \geq S'_j) = \begin{cases} 1, if \ m_i \geq m_j, \\ \frac{u_i - l_j}{(u_i - m_i) + (m_j - l_j)}, l_j \leq u_i, \ i, j = 1, \ldots, n; j \neq i \\ 0, \text{ otherwise} \end{cases} \quad (14)$$

where $S'_i = (l_i, m_i, u_i)$ and $S'_j = (l_j, m_j, u_j)$.
Considering that:

$$w'(a_i) = \min \{ V(S'_i \geq S'_k) \}, \ k = 1, 2, \ldots, n; k \neq i \quad (15)$$

the weight vector values can be calculated by:

$$w'(a_i) = [w'(a_1), \ w'(a_2), \ \ldots, \ w'(a_n)]^T. \quad (16)$$

The weight vectors were obtained using the following equation after a normalization process:

$$w(a_i) = [w(a_1), \ w(a_2), \ \ldots, \ w(a_n)]^T \quad (17)$$

where $w$ is a non-fuzzy number.

The present study was carried out by completing several methodological steps, as presented in Figure 5 and also briefly described below.

### 3.2. Data Used

#### 3.2.1. Torrential Areas Inventory

Identifying the areas previously affected by a natural risk phenomenon is vital for detecting other zones where that phenomenon has a high probability of occurrence [41]. The appearance of any phenomenon will be favored in areas with characteristics similar to those where the phenomenon has already occurred [42]. For this reason, to estimate the susceptibility to the occurrence of rapid floods, torrential areas were inventoried and mapped. These areas were generated by the rapid surface runoff on the slopes. The modality of identification of such zones is presented in the studies [43]. Torrential areas are zones characterized by the unified presence of a torrential microform of relief such as ravines and gullies generated by surface runoff. Thus, through the satellite images made available through the Google Earth application (Figure 1), an area affected by intense torrential processes of about 170 hectares was vectorized, which is located in the upper part of the river basin where the absence of vegetation and the high slopes favor the apparition of such phenomena.

#### 3.2.2. Flash-Flood and Flood Predictors

Whereas torrential zones represent an indicator of the rapid surface runoff on the slopes, certain geographical factors are the predictors of this phenomenon, or in other

words, are the variables that generate and favor the surface runoff. Moreover, the genesis of floods generated by flash-floods also depends on the characteristics of geographical factors. Therefore, to identify as accurately as possible the surfaces favorable to the genesis of flash-floods and those susceptible to floods, twelve conditioning factors were taken into account. Eight morphometrical predictors were obtained by processing the digital elevation model, while the other four flood and flash-flood predictors were collected from the following vector databases: hydrological soil groups from the Digital Soil Map of Romania, 1:200,000; land use/cover from Corine Land Cover, 2018; lithology from the Digital Geological Map of Romania, 1:200,000; and distance from rivers was estimated with the help of the river network in an Environmental Systems Research Institute (ESRI) shapefile format. Below, the main characteristics of flood and flash-flood predictors are briefly presented.

The slope is the geographic factor that has the biggest influence on both the potential for rapid surface runoff and the flood potential [24]. Surfaces with steep slopes cause rapid water drainage, while the flat surfaces lead to the water accumulation process [44]. In our case study, the sloping relief had values between 0.1° and 54.1° (Figure 2a). This interval was divided into six classes according to the literature [43].

Land use/cover is another predictor that influences both flash-floods and floods [45]. Lands covered with pastures or without vegetation will favor the appearance of rapid runoff on the slopes, while areas covered with forests are characterized by a lower potential for runoff and flooding [21]. In the study area, the grassland and the forest shared equally almost all of the territory (Figure 2b). Additionally, the presence of the built space in the lower part of the Izvorul Dorului River basin was observed.

Hydrological soil group has a high influence on the flood. Thus, the flooding phenomenon will likely be over the areas with soils with high clay content such as those in hydrological group D, while water infiltration will be more pronounced on soils with a sandy texture [46–48]. Within the study area, the largest surface was occupied by hydrological soil group A (Figure 2c).

Convergence index (CI) is a predictor obtained from the DEM whose values show the concentration degree of the drainage network. CI values close to −100 indicate a high density of the river network whereas positive CI values are associated with the interfluvial surfaces. In the study area, the CI values are situated in the range from −86 to 84 (Figure 2d). These were divided into five classes according to the literature [43].

Profile curvature is a predictor whose negative values show the surfaces that favor the accelerated surface runoff, while the decelerated runoff manifests itself on the surfaces with positive values. The information from the literature was used to classify profile curvature values into the next classes: −2.3––0.1; 0–0.1; 0.2–2.6 (Figure 3a).

The aspect factor obtained from the DEM is an indicator of the humidity potential that exists at the slope level [49]. In the case of the Izvorul Dorului basin, the southeast surfaces were the most extensive, these being followed by the southwest slopes (Figure 3b).

Topographic position index (TPI) is a predictor calculated from the DEM, which shows the relative position of a point in the research area in relation to the immediately neighboring regions [50]. The next TPI classes were established using the natural breaks method: −7.8––1.8; −1.7––0.5; −0.4–0.5; 0.6–1.9; 2–8.6 (Figure 3c). The following five classes of Topographic Wetness Index (TWI) were delimited using the natural break method: −4.4–4.7; 4.8–8.4; 8.5–11.8; 11.9–15; 15.1–23.1 (Figure 3d).

The elevation is a useful indicator for detecting the surfaces exposed to flooding processes that may occur as a result of flash-flood propagation from the upper part of river basins [7]. The lower relief zones have a higher sensitivity to flooding occurrence. For the study area, the range from 763.1 m to 2202 m was split into seven classes that generally succeeded at a difference of 200 m (Figure 4a).

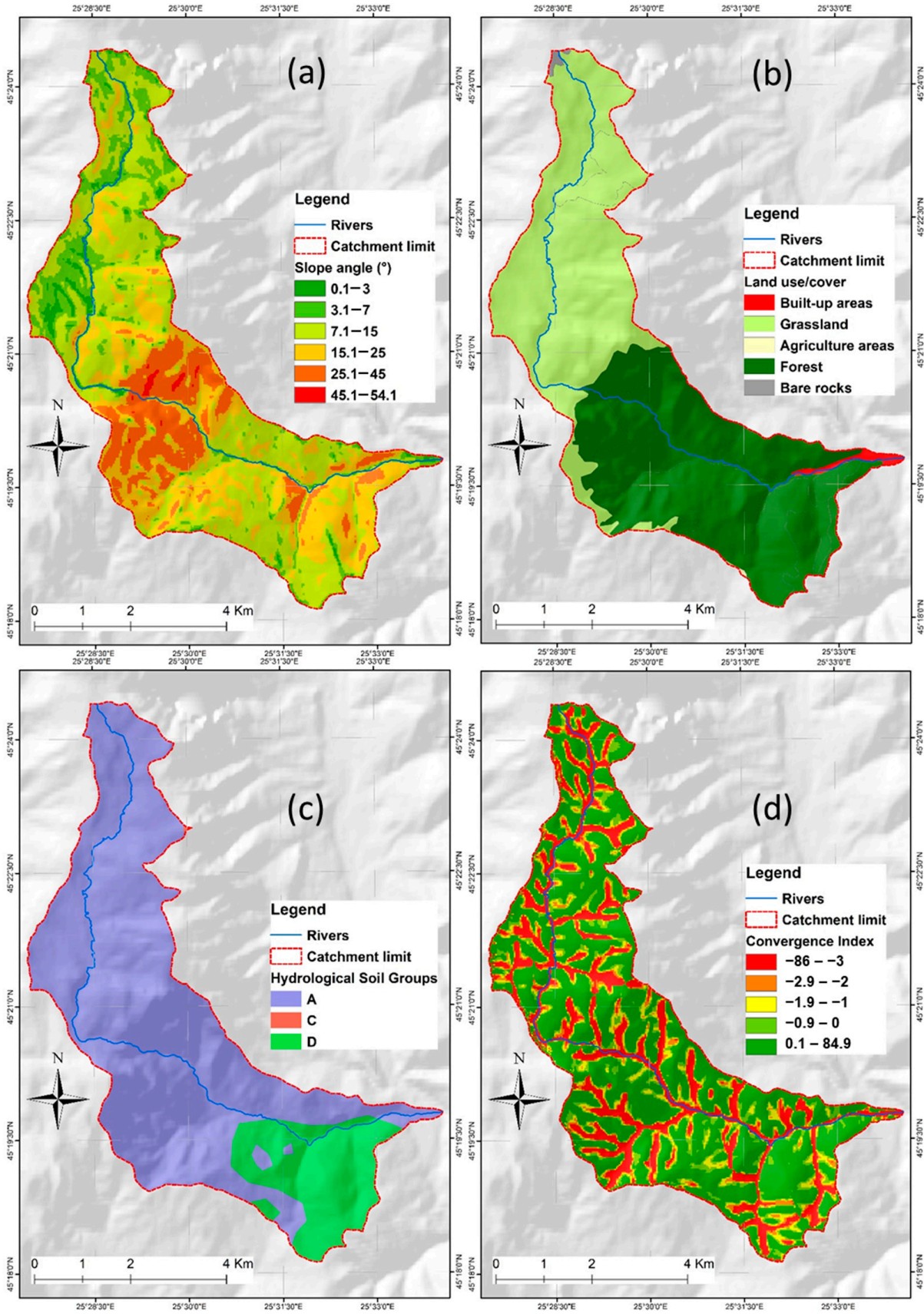

**Figure 2.** Flash-flood and flood predictors. (**a**) Slope; (**b**) Land use; (**c**) Hydrological Soil Group; (**d**) Convergence index.

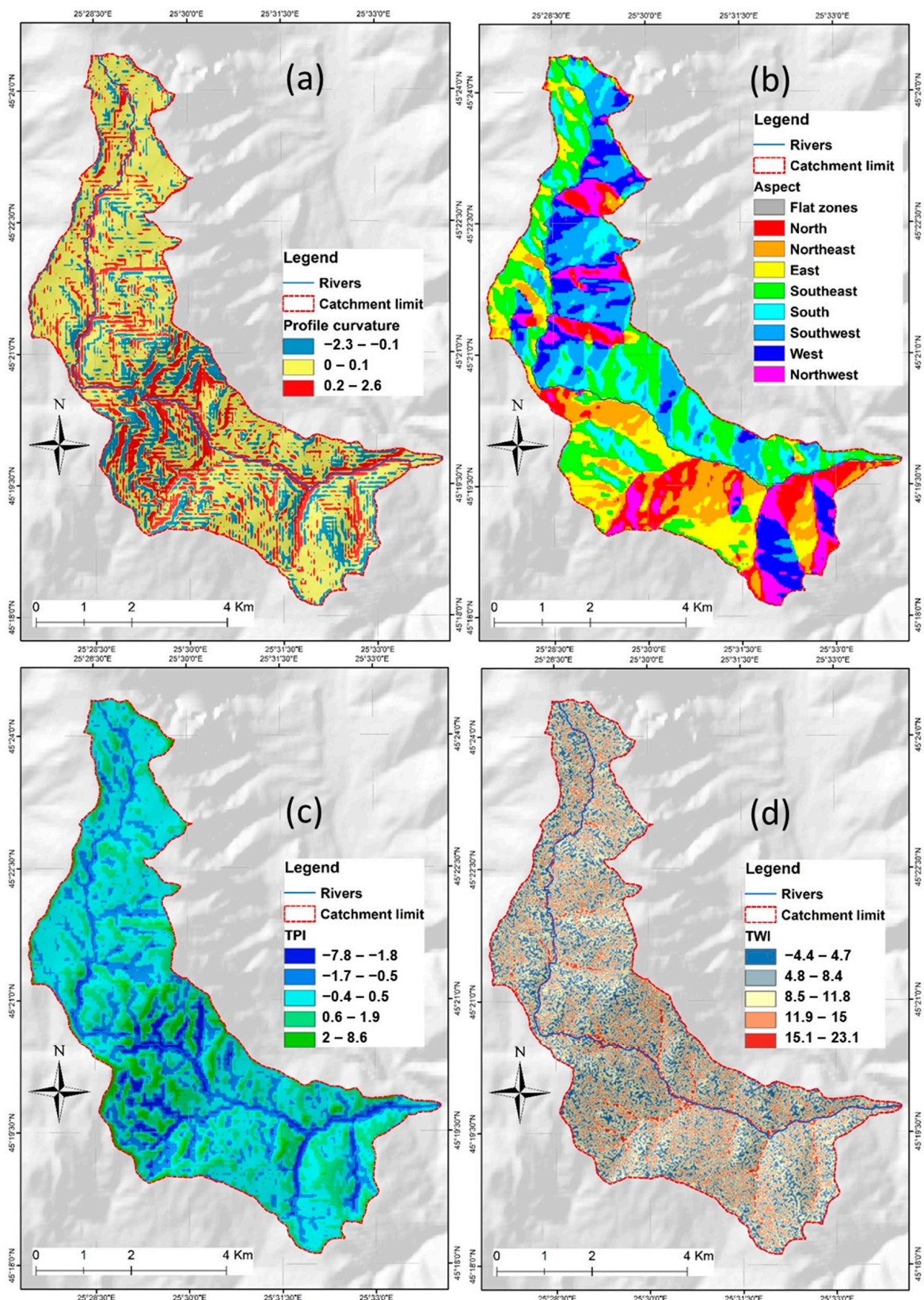

**Figure 3.** Flash-flood and flood predictors. (**a**) Profile curvature; (**b**) Aspect; (**c**) Topographic position index (TPI); (**d**) Topographic Wetness Index TWI.

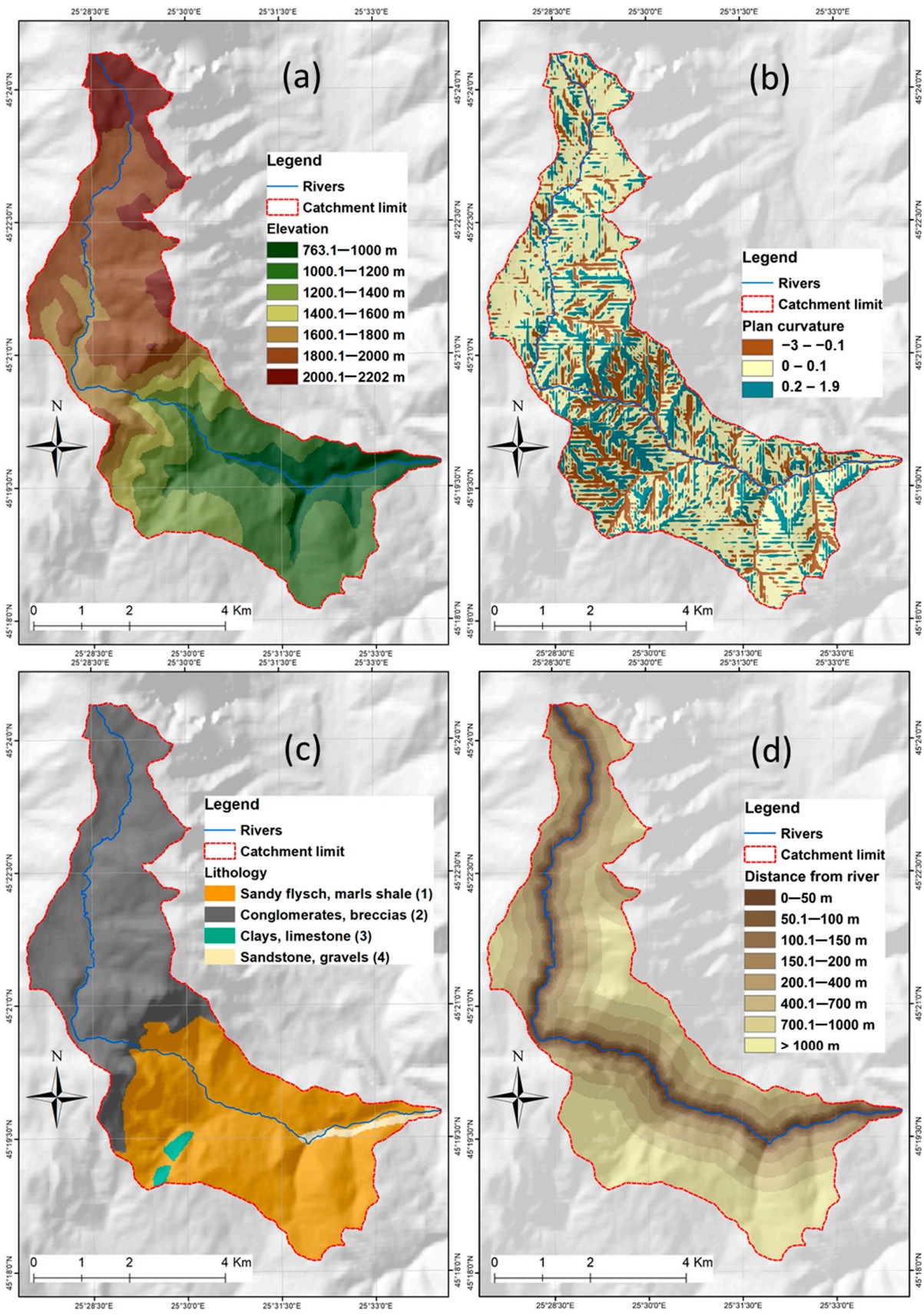

**Figure 4.** Flash-flood and flood predictors. (**a**). Elevation; (**b**) Plan curvature; (**c**) Lithology; (**d**) Distance from rive.

Plan curvature shows the difference between the surfaces on which the convergent and divergent runoff is manifested. Three classes were delimited for the plan curvature values (Figure 4b): $-3--0.1$; $0-0.1$; $0.2-1.9$.

Lithology is a predictor that influences the infiltration capacity at the ground surface, so it should be considered in the studies concerning the flood and flash-flood potential. The conglomerates, breccias, sandy flysch, and marls shale are predominant in the study area (Figure 4c).

Distance from the river was generated using the Euclidean distance tool from ArcGIS 10.3 software. This is an important parameter that indicates the distance from different surfaces to the nearest watercourse. The surfaces in the vicinity of watercourses will be more prone to flash-floods and the floods generated by them. In the present study, the distance from the river predictor was classified into eight classes.

### 3.3. Methodological Steps Implemented in the Present Study

### 3.3.1. Step 1: Flash-Flood Database Preparation

The flash-flood database used in the present research consisted of 1965 torrential points collected from the delineated torrential surfaces and ten flash-flood conditioning factors. Building and processing the flash-flood database were done through ArcGIS 10.3 software. It should be noted that the torrential points were obtained by converting the torrential areas from a raster format, with a cell size of 30 m, to a point. Therefore, each point corresponds to a raster cell. According to the literature [51], the entire sample was divided into a training dataset (70%) and a validation dataset (30%). The training dataset was used to calculate the frequency ratio, weights of evidence, and statistical index coefficients, while the validation dataset was used to evaluate the accuracy of the results achieved.

### 3.3.2. Step 2: Computation of Flash-Flood Potential Index (FFPI)

The flash-flood potential index represents a qualitative indicator of the potential for torrential surface runoff, which exists at the slope level [52]. In the first stage, the frequency ratio, weights of evidence, and statistical index coefficients were determined by analyzing the spatial correlations between the torrential points included in the training sample and the ten flash-flood predictors. In this regard, the equations from Sections 3.1.1–3.1.3 were implemented in Excel and ArcGIS. The number of pixels used in the computation of the types of bivariate statistics coefficients was 1376. Furthermore, the second stage consisted of the computation of flash-flood predictors weights by the fuzzy analytical hierarchy process method. Finally, three variants of the flash-flood potential index were computed by the weighted sum between fuzzy analytical hierarchy process weights and the values of frequency ratio, weights of evidence, and statistical index coefficients.

### 3.3.3. Step 3: Evaluation of Results Accuracy Using Receiver Operating Characteristic (ROC) Curve

The results of FFPI were assessed using the receiver operating characteristic (ROC) curve. The ROC curve represents a graphical plot that highlights the ability of a binary model to classify a given dataset used in the modeling process into the presence or the absence of a specific phenomenon [53]. This is the most frequently used algorithm to validate the outcomes provided by a model for natural hazards susceptibility [42,49,54,55]. The ROC curves were constructed by comparing the existing torrential points with the flash-flood potential index results. Both the success rate, constructed with the training sample, and prediction rate constructed with the validation sample, will be used. The area under curve (AUC) will highlight the performance of each flash-flood potential index model.

### 3.3.4. Step 4: Computation the Flood Potential Index (FPI) Based on the Most Performant FFPI Result

To identify the valleys with a high torrential degree, the best performing flash-flood potential index that resulted was used in a *flow accumulation* procedure (Figure 5). Through the *flow accumulation* method, the flash-flood potential index values are weighted at the

level of the river network within the study area. The weighted flash-flood potential index values are further classified into five categories: very low, low, medium, high, and very high. In the next stage, to select the river valleys along which the flood potential index will be calculated, the hydrographic network having assigned a medium, high, and very high flash-flood propagation susceptibility is selected. The flood potential index represents a qualitative indicator that highlights the degree to which a specific region can be affected by the flooding phenomenon [56]. The area on which the flood potential index will be computed was limited to a buffer zone of 200 m along with the selected river network. Eventually, the flood potential index values are obtained by involving the next ten flood conditioning factors in the analytical hierarchy process method: slope, land use, hydrological soil groups, convergence index, topographic position index, topographic wetness index, elevation, distance from the river, plan curvature, and lithology. The values of the FPI are then classified into five categories through which the areas prone to flooding generated by flash-floods will be detected.

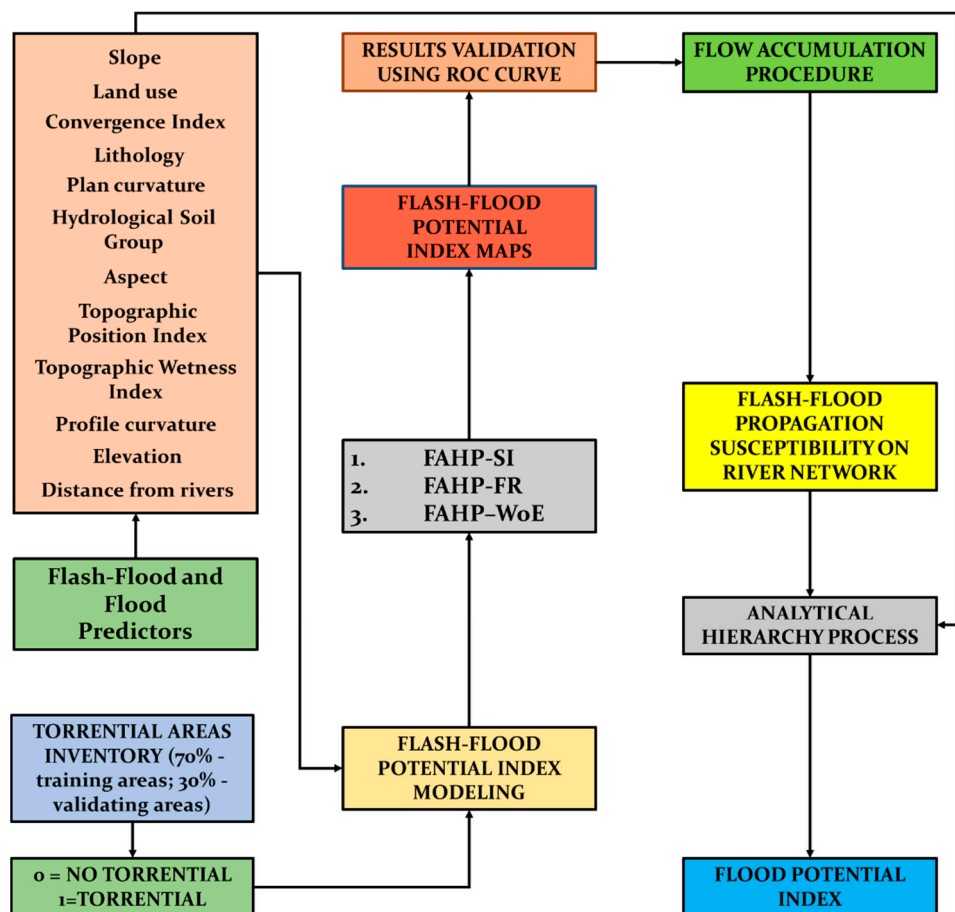

**Figure 5.** Flowchart of the present research.

## 4. Results

### 4.1. Bivariate Statistics Coefficients

The values of bivariate statistics coefficients highlight the spatial relationships between the location of torrential areas and the classes/categories of flash-flood predictors. According to Table 1, the lowest weights of evidence coefficients were assigned to hydrological soil group D (−15.04), lithological category of sandy flysch, marls shale (−10.3), lithological category of clays, limestone (−9.27), and hydrological soil group C (−8.91). The highest weights of evidence values were attributed to slope angles higher than 45° (3.34), grassland land use (1.52), lithological category of conglomerates, breccias (1.48),

and slope angles between 15.1° and 25° (0.95). In terms of frequency ratio coefficients, the lowest values (0) were associated with agricultural zones, built-up areas, bare rocks, lithological categories of sandstone, gravels, clays, and limestone, zones with slope angles lower than 3°, and hydrological soil groups C and D. The highest frequency ratio values were assigned to zones with slope angles higher than 45° (13.5), grassland land use (2.07), lithological category of conglomerates, breccias (2.04), areas with slope angles between 15.1° and 25° (1.82), and convergence index class between −86 and −3 (1.78). In the case of SI coefficients, the lowest values were calculated for hydrological soil group D (−7.83), zones with slopes lower than 3° (−5.99), lithological category of sandstone and gravels (−5.25), lithological category of sandy flysch and marls shale (−5.12), and built-up areas (−4.94). The highest SI coefficients were obtained by zones with slope angles higher than 45° (2.6), grassland land use (0.73), lithological category of conglomerates, breccias (0.71), zones with slope angles between 15.1° and 25° (0.6), and convergence index class between −86 and −3 (0.58).

**Table 1.** Bivariate statistics of flash-floods conditioning factors classes.

| Factor | Class | Class Pixels | Torrential Points | WOE | FR | SI |
|---|---|---|---|---|---|---|
| Slope | 0–3° | 1065 | 0 | −6.05 | 0.00 | −5.99 |
| | 3.1–7° | 5008 | 230 | 0.26 | 1.22 | 0.20 |
| | 7.1–15° | 15,098 | 299 | −0.94 | 0.53 | −0.64 |
| | 15.1–25° | 9937 | 677 | 1.01 | 1.82 | 0.60 |
| | 25.1–45° | 5407 | 93 | −0.88 | 0.46 | −0.78 |
| | 45.1–54° | 152 | 77 | 3.34 | 13.50 | 2.60 |
| Land use | Built-up areas | 374 | 0 | −6.78 | 0 | −4.94 |
| | Grassland | 16,949 | 1316 | 1.52 | 2.07 | 0.73 |
| | Agriculture areas | 15 | 0 | −3.55 | 0 | −1.73 |
| | Forest | 19,218 | 60 | −5.05 | 0.08 | −2.49 |
| | Bare rocks | 111 | 0 | −5.56 | 0 | −3.73 |
| Convergence index | −86–−3 | 7514 | 502 | 0.93 | 1.78 | 0.58 |
| | −2.9–−2 | 2085 | 114 | 0.51 | 1.46 | 0.38 |
| | −1.9–−1 | 2750 | 107 | 0.13 | 1.04 | 0.04 |
| | −0.9–0 | 3939 | 101 | −0.35 | 0.68 | −0.38 |
| | 0.1–84.9 | 20,379 | 552 | −0.56 | 0.72 | −0.33 |
| Lithology | Sandy flysch, marls shale | 17,891 | 4 | −10.30 | 0.01 | −5.12 |
| | Conglomerates, breccias | 17,948 | 1372 | 1.48 | 2.04 | 0.71 |
| | Clays, limestone | 321 | 0 | −9.27 | 0 | −4.79 |
| | Sandstone, gravels | 507 | 0 | −4.44 | 0 | −5.25 |
| Plan curvature | −3–−0.1 | 7202 | 345 | 0.29 | 1.28 | 0.244 |
| | 0–0.1 | 20,963 | 821 | 0.07 | 1.04 | 0.043 |
| | 0.2–1.9 | 8502 | 210 | −0.57 | 0.66 | −0.418 |
| HSG | A | 29,965 | 1376 | 0.42 | 1.22 | 0.20 |
| | C | 18 | 0 | −8.91 | 0 | −1.91 |
| | D | 6684 | 0 | −15.04 | 0 | −7.83 |
| Aspect | Flat surfaces | 93 | 2 | −0.31 | 0.57 | −0.56 |
| | North | 3306 | 97 | −0.01 | 0.78 | −0.25 |
| | Northeast | 4440 | 27 | −1.70 | 0.16 | −1.82 |
| | East | 5064 | 104 | −0.43 | 0.55 | −0.60 |
| | Southeast | 6455 | 181 | −0.09 | 0.75 | −0.29 |
| | South | 5225 | 334 | 0.95 | 1.70 | 0.53 |
| | Southwest | 5434 | 286 | 0.69 | 1.40 | 0.34 |
| | West | 3829 | 211 | 0.73 | 1.47 | 0.38 |
| | Northwest | 2821 | 134 | 0.53 | 1.27 | 0.24 |
| TPI | −7.8–−1.8 | 2063 | 27 | −1.21 | 0.35 | −1.05 |
| | −1.7–−0.5 | 8121 | 380 | 0.22 | 1.25 | 0.22 |
| | −0.4–0.5 | 16,532 | 744 | 0.29 | 1.20 | 0.18 |
| | 0.6–1.9 | 8181 | 192 | −0.68 | 0.63 | −0.47 |
| | 2–8.6 | 1770 | 33 | −0.83 | 0.50 | −0.70 |
| TWI | −4.4–4.7 | 7477 | 277 | −0.03 | 0.99 | −0.01 |
| | 4.8–8.4 | 9509 | 376 | 0.06 | 1.05 | 0.05 |
| | 8.5–11.8 | 9180 | 307 | −0.17 | 0.89 | −0.12 |
| | 11.9–15 | 9414 | 404 | 0.18 | 1.14 | 0.13 |
| | 15.1–23.1 | 1083 | 12 | −1.28 | 0.30 | −1.22 |
| Profile curvature | −3–−0.1 | 7255 | 185 | −0.65 | 0.68 | −0.39 |
| | 0–0.1 | 21,678 | 957 | 0.30 | 1.18 | 0.16 |
| | 0.2–1.9 | 7734 | 234 | −0.45 | 0.81 | −0.22 |

*4.2. Flash-Flood Potential Index Computation Using Fuzzy Analytical Hierarchy Process Based Ensembles*

Following the methodological steps described in Section 3.1.4 the fuzzy analytical hierarchy process algorithm was applied to determine the weights of the flash-flood predictors. In the first step, the fuzzy analytical hierarchy process evaluation matrix was created based on expert judgment (Table 2). Furthermore, using the values included in the evaluation matrix, the synthesis values were calculated by using the formula:

$$\left[\sum_{k=1}^{n}\sum_{j=1}^{n} a'_{kj}\right]^{-1} = (88.48\ 130.16\ 182.17)^{-1} = (0.005\ 0.008\ 0.011) \tag{18}$$

**Table 2.** Fuzzy analytical hierarchy process evaluation matrix.

|  | 1 | 2 | 3 | 4 | 5 | 6 | 7 | 8 | 9 | 10 |
|---|---|---|---|---|---|---|---|---|---|---|
| Slope (1) | | | | | | | | | | |
| $l_1$ | 1 | 1 | 2 | 1 | 1 | 2 | 3 | 3 | 2 | 1 |
| $m_1$ | 1 | 2 | 3 | 2 | 2 | 3 | 4 | 4 | 3 | 2 |
| $u_1$ | 1 | 3 | 4 | 3 | 3 | 4 | 5 | 5 | 4 | 3 |
| Land use (2) | | | | | | | | | | |
| $l_2$ | 0.33 | 1 | 1 | 1 | 1 | 1 | 2 | 1 | 1 | 1 |
| $m_2$ | 0.5 | 1 | 2 | 1 | 2 | 2 | 3 | 2 | 2 | 1 |
| $u_2$ | 1 | 1 | 3 | 1 | 3 | 3 | 4 | 3 | 3 | 1 |
| Convergence index (3) | | | | | | | | | | |
| $l_3$ | 0.25 | 0.33 | 1 | 0.33 | 0.33 | 1 | 1 | 1 | 1 | 0.33 |
| $m_3$ | 0.33 | 0.5 | 1 | 0.5 | 0.5 | 1 | 2 | 2 | 1 | 0.5 |
| $u_3$ | 0.5 | 1 | 1 | 1 | 1 | 1 | 3 | 3 | 1 | 1 |
| Lithology (4) | | | | | | | | | | |
| $l_4$ | 0.33 | 1 | 1 | 1 | 1 | 1 | 2 | 2 | 1 | 1 |
| $m_4$ | 0.5 | 1 | 2 | 1 | 1 | 2 | 3 | 3 | 2 | 1 |
| $u_4$ | 1 | 1 | 3 | 1 | 1 | 3 | 4 | 4 | 3 | 1 |
| Plan curvature (5) | | | | | | | | | | |
| $l_5$ | 0.33 | 1 | 1 | 1 | 1 | 1 | 2 | 2 | 1 | 1 |
| $m_5$ | 0.5 | 1 | 2 | 1 | 1 | 2 | 3 | 3 | 2 | 1 |
| $u_5$ | 1 | 1 | 3 | 1 | 1 | 3 | 4 | 4 | 3 | 1 |
| HGS (6) | | | | | | | | | | |
| $l_6$ | 0.25 | 0.33 | 1 | 0.33 | 0.33 | 1 | 1 | 1 | 1 | 0.33 |
| $m_6$ | 0.33 | 0.5 | 1 | 0.5 | 0.5 | 1 | 2 | 2 | 1 | 0.5 |
| $u_6$ | 0.5 | 1 | 1 | 1 | 1 | 1 | 3 | 3 | 1 | 1 |

**Table 2.** *Cont.*

|  | 1 | 2 | 3 | 4 | 5 | 6 | 7 | 8 | 9 | 10 |
|---|---|---|---|---|---|---|---|---|---|---|
| Aspect (7) | | | | | | | | | | |
| $l_7$ | 0.2 | 0.25 | 0.33 | 0.25 | 0.25 | 0.33 | 1 | 1 | 1 | 0.33 |
| $m_7$ | 0.25 | 0.33 | 0.5 | 0.33 | 0.33 | 0.5 | 1 | 1 | 1 | 0.5 |
| $u_7$ | 0.33 | 0.5 | 1 | 0.5 | 0.5 | 1 | 1 | 1 | 1 | 1 |
| TPI (8) | | | | | | | | | | |
| $l_8$ | 0.2 | 0.25 | 0.33 | 0.25 | 0.25 | 0.33 | 1 | 1 | 1 | 0.33 |
| $m_8$ | 0.25 | 0.33 | 0.5 | 0.33 | 0.33 | 0.5 | 1 | 1 | 1 | 0.5 |
| $u_8$ | 0.33 | 0.5 | 1 | 0.5 | 0.5 | 1 | 1 | 1 | 1 | 1 |
| TWI (9) | | | | | | | | | | |
| $l_9$ | 0.25 | 0.33 | 1 | 0.33 | 0.33 | 1 | 1 | 1 | 1 | 0.33 |
| $m_9$ | 0.33 | 0.5 | 1 | 0.5 | 0.5 | 1 | 2 | 2 | 1 | 0.5 |
| $u_9$ | 0.5 | 1 | 1 | 1 | 1 | 1 | 3 | 3 | 1 | 1 |
| Profile curvature (10) | | | | | | | | | | |
| $l_{10}$ | 0.33 | 1 | 1 | 1 | 1 | 1 | 2 | 1 | 1 | 1 |
| $m_{10}$ | 0.5 | 1 | 2 | 1 | 2 | 2 | 3 | 2 | 2 | 1 |
| $u_{10}$ | 1 | 1 | 3 | 1 | 3 | 3 | 4 | 3 | 3 | 1 |

The synthesis values calculated above were used in the following step to calculate the fuzzy numbers for each flash-flood predictor. The fuzzy numbers were then compared using the degree of possibility procedure, which is exemplified in Table 3. Utilizing the results provided by the degree of possibility method, the weight vector values were calculated using the following relations:

$$w'(a_i) = \{1\ 0.71\ 0.32\ 0.68\ 0.68\ 0.32\ 0\ 0\ 0.32\ 0.71\}^T \tag{19}$$

$$w(a_i) = \{0.211\ 0.15\ 0.066\ 0.143\ 0.143\ 0.066\ 0\ 0\ 0.066\ 0.15\}^T \tag{20}$$

In the next step, employing the defuzzification procedure, the Triangular Fuzzy Numbers (TFNs) were transformed into the crisp weights that will be attributed to each flash-flood predictor and multiplied with statistical index, frequency ratio, and weights of evidence values to obtain the flash-flood potential index.

Flash-flood potential index values were mapped using the map algebra capability from ArcGIS software. All three flash-flood potential indices were standardized between 0 and 1 and then reclassified into five classes using the natural break method. In the case of FFPI$_{FAHP-SI}$, very low values, situated from 0 to 0.25, were found in about 2.82% of the study area (Figure 6a). The values, ranging from 0.26 to 0.62, were mainly associated with the southern half and represent 28.9% of the entire river basin. The medium FFPI$_{FAHP-SI}$ class corresponded to approximately 13.86% of the Izvorul Dorului catchment. The high and very high potential were spread over a total of 54.43% of the entire catchment surface. The analysis of FFPI$_{FAHP-FR}$ revealed that the very low potential spanned 18.48% of the total study area and was present mainly in the southern half. The low flash-flood potential accounted for approximately 16.29% of the catchment surface, while the medium FFPI$_{FAHP-WOE}$, with values from 0.32 to 0.48, covered 26.66% of the study zone. A zone including 38.57% of the research area was characterized by a high and very high flash-flood potential (Figure 6b). Following the application of the FAHP-WOE ensemble, only 0.62% of the Izvorul Dorului catchment had a very low flash-flood potential (Figure 6c). The low flash-flood potential, with values between 0.26 and 0.44, covered around 10.62% of

the entire territory, while the medium values quantified approximately 30.81% of the river basin. A percentage of 57.95% of the study area had high and very high FFPI$_{\text{FAHP-WOE}}$ values ranging from 0.64 to 1.

**Table 3.** The ordinate of the highest intersection point, the degree possibility for Triangular Fuzzy Numbers (TFNs), and the weights of the flash-flood predictors.

| Slope = 1 | Land Use = 2 | CI = 3 | Lithology = 4 | Plan Curvature = 5 |
|---|---|---|---|---|
| V(S1 ≥ S2) = 1 | V(S2 ≥ S1) = 0.71 | V(S3 ≥ S1) = 0.32 | V(S4 ≥ S1) = 0.68 | V(S5 ≥ S1) = 0.68 |
| V(S1 ≥ S3) = 1 | V(S2 ≥ S3) = 1 | V(S3 ≥ S2) = 0.65 | V(S4 ≥ S2) = 1 | V(S5 ≥ S2) = 1 |
| V(S1 ≥ S4) = 1 | V(S2 ≥ S4) = 1 | V(S3 ≥ S4) = 0.62 | V(S4 ≥ S3) = 1 | V(S5 ≥ S3) = 1 |
| V(S1 ≥ S5) = 1 | V(S2 ≥ S5) = 1 | V(S3 ≥ S5) = 0.62 | V(S4 ≥ S5) = 1 | V(S5 ≥ S4) = 1 |
| V(S1 ≥ S6) = 1 | V(S2 ≥ S6) = 1 | V(S3 ≥ S6) = 1 | V(S4 ≥ S6) = 1 | V(S5 ≥ S6) = 1 |
| V(S1 ≥ S7) = 1 | V(S2 ≥ S7) = 1 | V(S3 ≥ S7) = 1 | V(S4 ≥ S7) = 1 | V(S5 ≥ S7) = 1 |
| V(S1 ≥ S8) = 1 | V(S2 ≥ S8) = 1 | V(S3 ≥ S8) = 1 | V(S4 ≥ S8) = 1 | V(S5 ≥ S8) = 1 |
| V(S1 ≥ S9) = 1 | V(S2 ≥ S9) = 1 | V(S3 ≥ S9) = 1 | V(S4 ≥ S9) = 1 | V(S5 ≥ S9) = 1 |
| V(S1 ≥ S10) = 1 | V(S2 ≥ S10) = 1 | V(S3 ≥ S10) = 0.65 | V(S4 ≥ S10) = 1 | V(S5 ≥ S10) = 1 |
| min{V(S$_1$ ≥ S$_k$)} = 1 | min{V(S$_2$ ≥ S$_k$)} = 0.71 | min{V(S$_3$ ≥ S$_k$)} = 0.32 | min{V(S$_4$ ≥ S$_k$)} = 0.68 | min{V(S$_5$ ≥ S$_k$)} = 0.68 |
| Weight = 0.211 | Weight = 0.15 | Weight = 0.066 | Weight = 0.143 | Weight = 0.143 |
| **HSG = 6** | **Aspect = 6** | **TPI = 7** | **TWI = 8** | **Profile Curvature = 10** |
| V(S6 ≥ S1) = 0.32 | V(S7 ≥ S1) = 0 | V(S8 ≥ S1) = 0 | V(S9 ≥ S1) = 0.32 | V(S10 ≥ S1) = 0.71 |
| V(S6 ≥ S2) = 0.65 | V(S7 ≥ S2) = 0.23 | V(S8 ≥ S2) = 0.23 | V(S9 ≥ S2) = 0.65 | V(S10 ≥ S2) = 1 |
| V(S6 ≥ S3) = 1 | V(S7 ≥ S3) = 0.59 | V(S8 ≥ S3) = 0.59 | V(S9 ≥ S3) = 1 | V(S10 ≥ S3) = 1 |
| V(S6 ≥ S4) = 0.62 | V(S7 ≥ S4) = 0.19 | V(S8 ≥ S4) = 0.19 | V(S9 ≥ S4) = 0.62 | V(S10 ≥ S4) = 1 |
| V(S6 ≥ S5) = 0.62 | V(S7 ≥ S5) = 0.19 | V(S8 ≥ S5) = 0.19 | V(S9 ≥ S5) = 0.62 | V(S10 ≥ S5) = 1 |
| V(S6 ≥ S7) = 1 | V(S7 ≥ S6) = 0.59 | V(S8 ≥ S6) = 0.59 | V(S9 ≥ S6) = 1 | V(S10 ≥ S6) = 1 |
| V(S6 ≥ S8) = 1 | V(S7 ≥ S8) = 1 | V(S8 ≥ S7) = 1 | V(S9 ≥ S7) = 1 | V(S10 ≥ S7) = 1 |
| V(S6 ≥ S9) = 1 | V(S7 ≥ S9) = 0.59 | V(S8 ≥ S9) = 0.59 | V(S9 ≥ S8) = 1 | V(S10 ≥ S8) = 1 |
| V(S6 ≥ S10) = 0.63 | V(S7 ≥ S10) = 0.23 | V(S8 ≥ S10) = 0.23 | V(S9 ≥ S10) = 0.63 | V(S10 ≥ S9) = 1 |
| min{V(S$_6$ ≥ S$_k$)} = 0.32 | min{V(S$_7$ ≥ S$_k$)} = 0 | min{V(S$_8$ ≥ S$_k$)} = 0 | min{V(S$_9$ ≥ S$_k$)} = 0.32 | min{V(S$_{10}$ ≥ S$_k$)} = 0.71 |
| Weight = 0.066 | Weight = 0 | Weight = 0 | Weight = 0.066 | Weight = 0.15 |

### 4.3. Flash-Flood Potential Index Results Validation

Results validation is a mandatory step to establish the best ensemble model whose results will be used to identify the areas prone to flood generated by flash-floods. In this regard, the success rate and prediction rate were used. The success rate revealed that the highest performance was obtained by the results provided by FAHP-WOE (AUC = 0.837), followed by FAHP-SI (AUC = 0.833) and FAHP-FR (AUC = 0.723) (Figure 7a). The same hierarchy was also revealed by the construction of the prediction rate. Thus, the FAHP-WOE ranked first (AUC = 0.79), followed by FAHP-SI (AUC = 0.787) and FAHP-FR (AUC = 0.717) (Figure 7b). Therefore, following the results validation procedure, the FFPI$_{\text{FAHP-WOE}}$ was selected to be used in the next step of the analysis.

### 4.4. Flood Potential Index Computation

According to the description provided in Section 3.3.4, the flow accumulation method was applied to FFPI$_{\text{FAHP-WOE}}$ to evaluate the torrential degree of the river valleys across the study area. The results showed that a percentage of 21.59% of the total river valleys identified were characterized by a low and very low torrential degree and are, therefore, considered to be less favorable for flash-flood propagation (Figure 8a). For a 200 m buffer zone along with the other 78.41% of the river valleys, the flood potential index (FPI) was calculated. In this regard, the stand-alone analytical hierarchy process (AHP) multicriteria decision-making was used. It should be mentioned that through AHP, in the first stage, the weights of flash-flood predictors and classes/categories of flash-flood predictors were calculated. In terms of flash-flood predictors, the highest weight was detected for slope (0.224), followed by land use (0.137), elevation (0.137), distance from

river (0.137), lithology (0.085), plan curvature (0.081), hydrological soil groups (0.064), convergence index (0.055), TPI (0.046), and TWI (0.031) (Table 4). The analysis of the weights attributed to the classes/categories of flash-flood predictors revealed that the highest value was obtained for hydrological soil group D (0.66), followed by the plan curvature class between −3 and −0.1 (0.539), the TPI class between −7.8 and −1.8 (0.439), the TWI class between −4.4 and 4.7 (0.433), the conglomerates and breccias lithological categories (0.423), the convergence index class between −86 and −3 (0.42), and the slope angle class lower than 3° (0.379).

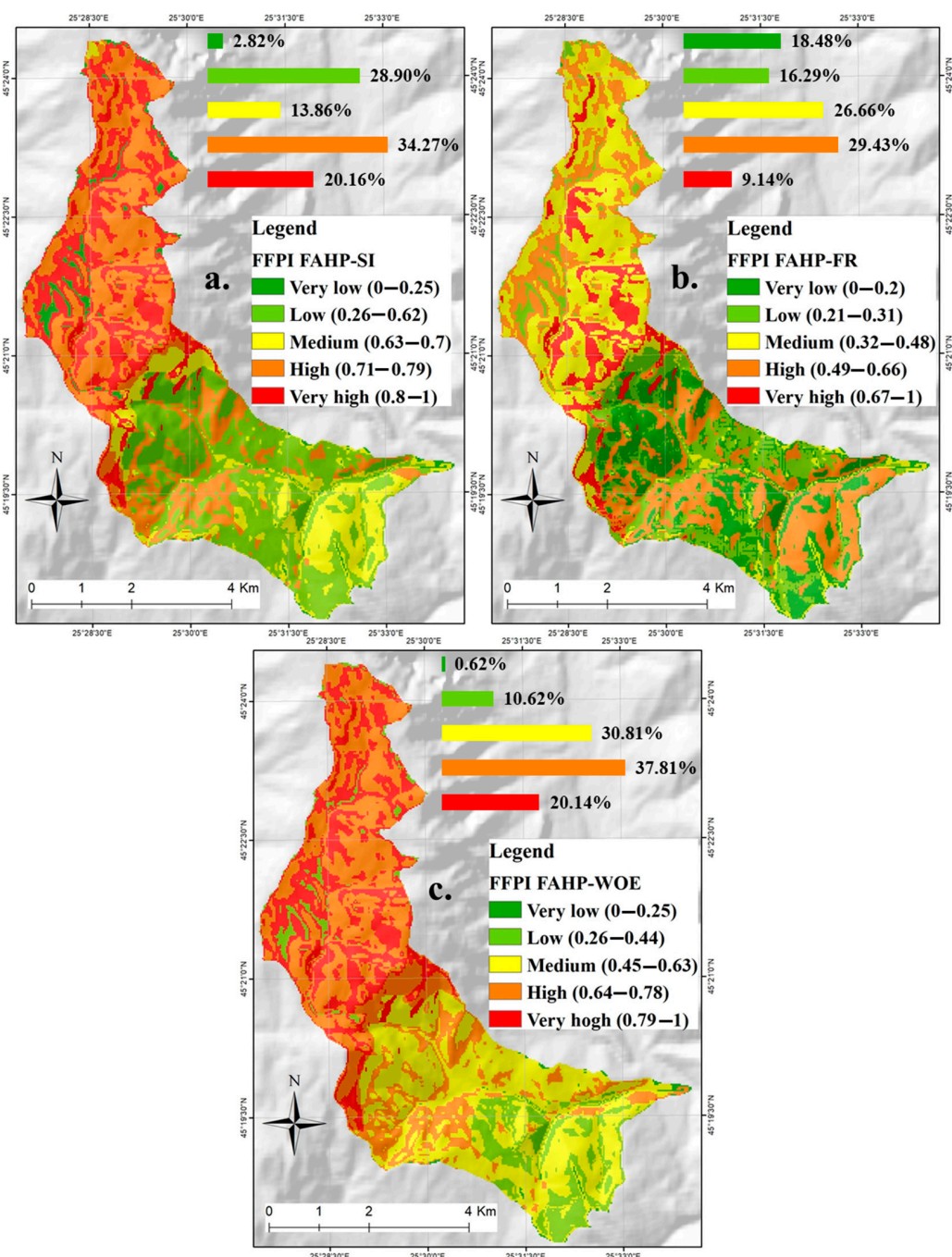

**Figure 6.** Flash-flood potential index (FFPI) values. (**a**) Fuzzy Analytical Hierarchy Process—Statistical Index (FAHP-SI); (**b**) Fuzzy Analytical Hierarchy Process—Frequency Ratio (FAHP-FR); (**c**) Fuzzy Analytical Hierarchy Process—Weights of Evidence (FAHP-WOE).

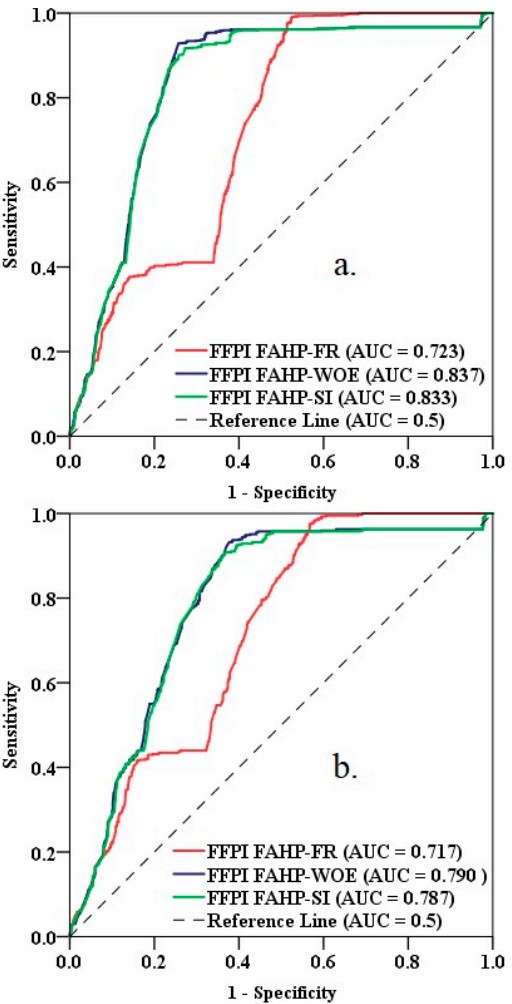

**Figure 7.** Receiver operating characteristic (ROC) Curves. (**a**) Success rate; (**b**) Prediction rate.

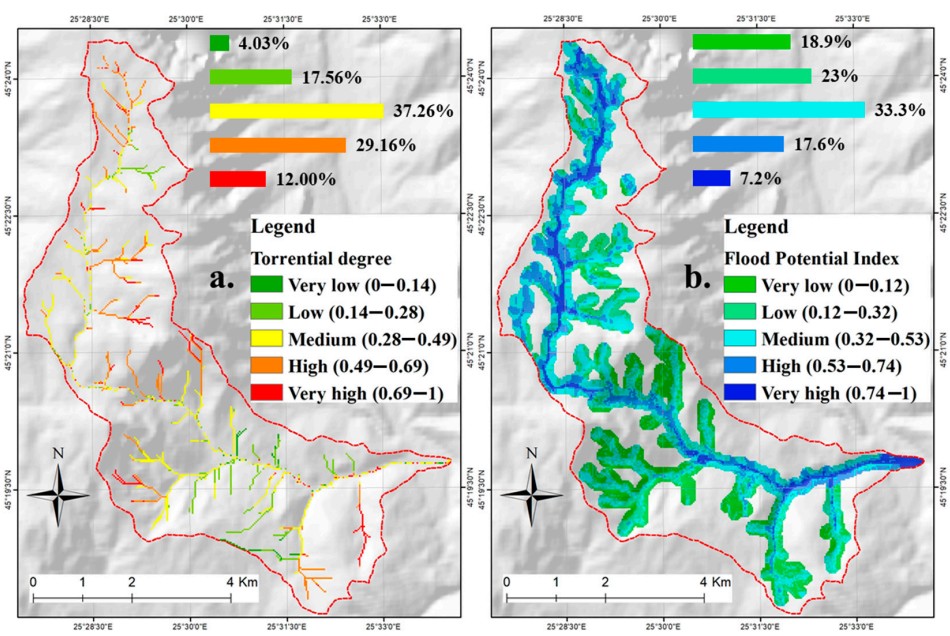

**Figure 8.** (**a**) River valleys torrential degree of the river; (**b**) Flood potential index classes.

**Table 4.** Pair-wise comparison matrix and normalized weights for each factor and class/category.

| Factor and Classes/Categories Flood Predictors | Pair-Wise Comparison Matrix [1] | [2] | [3] | [4] | [5] | [6] | [7] | [8] | [9] | [10] | AHP Weights |
|---|---|---|---|---|---|---|---|---|---|---|---|
| [1] Slope | 1 | | | | | | | | | | 0.224 |
| [2] TPI | 1/4 | 1 | | | | | | | | | 0.046 |
| [3] TWI | 1/5 | 1/2 | 1 | | | | | | | | 0.031 |
| [4] Land use | 1/2 | 3 | 4 | 1 | | | | | | | 0.137 |
| [5] Lithology | 1/3 | 2 | 3 | 1/2 | 1 | | | | | | 0.085 |
| [6] Elevation | 1/2 | 3 | 4 | 1 | 2 | 1 | | | | | 0.137 |
| [7] Distance from river | 1/2 | 3 | 4 | 1 | 2 | 1 | 1 | | | | 0.137 |
| [8] Plan curvature | 1/3 | 2 | 2 | 1/2 | 1 | 1/2 | 1/2 | 1 | | | 0.081 |
| [9] CI | 1/4 | 1 | 2 | 1/3 | 1/2 | 1/3 | 1/3 | 1/2 | 1 | | 0.055 |
| [10] HSG | 1/3 | 2 | 3 | 1/2 | 1/2 | 1/2 | 1/2 | 1/2 | 1/2 | 1 | 0.064 |
| *Classes in each factor* | | | | | | | | | | | |
| *Slope angle* | | | | | | | | | | | |
| [1] 0–3° | 1 | | | | | | | | | | 0.379 |
| [2] 3.1–7° | 1/2 | 1 | | | | | | | | | 0.249 |
| [3] 7.1–15° | 1/3 | 1/2 | 1 | | | | | | | | 0.160 |
| [4] 15.1–25° | 1/4 | 1/3 | 1/2 | 1 | | | | | | | 0.102 |
| [5] 25.1–45° | 1/5 | 1/4 | 1/3 | 1/2 | 1 | | | | | | 0.065 |
| [6] 45.1–54° | 1/6 | 1/5 | 1/4 | 1/3 | 1/2 | 1 | | | | | 0.043 |
| *TPI* | | | | | | | | | | | |
| [1] −7.8––1.8 | 1 | | | | | | | | | | 0.439 |
| [2] −1.7––0.5 | 1/2 | 1 | | | | | | | | | 0.255 |
| [3] −0.4–0.5 | 1/3 | 1/2 | 1 | | | | | | | | 0.156 |
| [4] 0.6–1.9 | 1/5 | 1/3 | 1/2 | 1 | | | | | | | 0.092 |
| [5] 2–8.6 | 1/6 | 1/4 | 1/3 | 1/2 | 1 | | | | | | 0.058 |
| *TWI* | | | | | | | | | | | |
| [1] −4.4–4.7 | 1 | | | | | | | | | | 0.433 |
| [2] 4.8–8.4 | 1/2 | 1 | | | | | | | | | 0.251 |
| [3] 8.5–11.8 | 1/3 | 1/2 | 1 | | | | | | | | 0.164 |
| [4] 11.9–15 | 1/5 | 1/3 | 1/2 | 1 | | | | | | | 0.100 |
| [5] 15.1–23.1 | 1/6 | 1/4 | 1/3 | 1/3 | 1 | | | | | | 0.052 |
| *Land use* | | | | | | | | | | | |
| [1] Built-up areas | 1 | | | | | | | | | | 0.328 |
| [2] Grassland | 1/2 | 1 | | | | | | | | | 0.189 |
| [3] Agriculture areas | 1/3 | 1/2 | 1 | | | | | | | | 0.120 |
| [4] Forest | 1/8 | 1/6 | 1/5 | 1 | | | | | | | 0.034 |
| [5] Bare rocks | 1 | 2 | 3 | 8 | 1 | | | | | | 0.328 |
| *Lithology* | | | | | | | | | | | |
| [1] Sandy flysch, marls shale | 1 | | | | | | | | | | 0.227 |
| [2] Conglomerates, breccias | 2 | 1 | | | | | | | | | 0.423 |
| [3] Clays, limestone | 1/2 | 1/3 | 1 | | | | | | | | 0.123 |
| [4] Sandstone, gravels | 1 | 1/2 | 2 | 1 | | | | | | | 0.227 |
| *Plan curvature* | | | | | | | | | | | |
| [1] −3––0.1 | 1 | | | | | | | | | | 0.539 |
| [2] 0–0.1 | 1/2 | 1 | | | | | | | | | 0.297 |
| [3] 0.2–1.9 | 1/3 | 1/2 | 1 | | | | | | | | 0.164 |
| *Elevation* | | | | | | | | | | | |
| [1] 763.1–1000 m | 1 | | | | | | | | | | 0.350 |
| [2] 1000.1–1200 m | 1/2 | 1 | | | | | | | | | 0.237 |
| [3] 1200.1–1400 m | 1/3 | 1/2 | 1 | | | | | | | | 0.159 |
| [4] 1400.1–1600 m | 1/4 | 1/3 | 1/2 | 1 | | | | | | | 0.107 |
| [5] 1600.1–1800 m | 1/5 | 1/4 | 1/3 | 1/2 | 1 | | | | | | 0.071 |
| [6] 1800.1–2000 m | 1/6 | 1/5 | 1/4 | 1/3 | 1/2 | 1 | | | | | 0.049 |
| [7] 2000.1–2202 m | 1/8 | 1/7 | 1/6 | 1/5 | 1/4 | 1/3 | 1 | | | | 0.026 |
| *Distance from river* | | | | | | | | | | | |
| [1] 0–50 m | 1 | | | | | | | | | | 0.327 |
| [2] 50.1–100 m | 1/2 | 1 | | | | | | | | | 0.227 |
| [3] 100.1–150 m | 1/3 | 1/2 | 1 | | | | | | | | 0.157 |
| [4] 150.1–200 m | 1/4 | 1/3 | 1/2 | 1 | | | | | | | 0.108 |
| [5] 200.1–400 m | 1/5 | 1/4 | 1/3 | 1/2 | 1 | | | | | | 0.073 |
| [6] 400.1–700 m | 1/6 | 1/5 | 1/4 | 1/3 | 1/2 | 1 | | | | | 0.050 |
| [7] 700.1–1000 m | 1/7 | 1/6 | 1/5 | 1/4 | 1/3 | 1/2 | 1 | | | | 0.034 |
| [8] >1000 m | 1/8 | 1/7 | 1/6 | 1/5 | 1/4 | 1/3 | 1/2 | 1 | | | 0.024 |
| *Convergence index* | | | | | | | | | | | |
| [1] −86––3 | 1 | | | | | | | | | | 0.420 |
| [2] −2.9––2 | 1/2 | 1 | | | | | | | | | 0.299 |
| [3] −1.9––1 | 1/3 | 1/3 | 1 | | | | | | | | 0.141 |
| [4] −0.9–0 | 1/4 | 1/4 | 1/2 | 1 | | | | | | | 0.088 |
| [5] 0.1–84.9 | 1/7 | 1/5 | 1/3 | 1/2 | 1 | | | | | | 0.052 |
| *HSG* | | | | | | | | | | | |
| [1] A | 1 | | | | | | | | | | 0.117 |
| [2] C | 3 | 1 | | | | | | | | | 0.224 |
| [3] D | 4 | 5 | 1 | | | | | | | | 0.660 |

The consistency of judgments was evaluated using the consistency ratio (CR) values. The results from Table 5 show that the CR values were less than 0.1, indicating that all the comparisons within the matrices were consistent. Table 5 also contains the values of some parameters involved in the CR computation.

**Table 5.** Properties of comparison matrices in the previous table.

| Factors | N | $\lambda_{max}$ | CI | RI | CR |
|---|---|---|---|---|---|
| All | 10 | 10.32 | 0.036 | 1.49 | 0.024 |
| Slope | 6 | 6.123 | 0.025 | 1.24 | 0.020 |
| TPI | 5 | 5.046 | 0.012 | 1.12 | 0.010 |
| TWI | 5 | 5.121 | 0.030 | 1.12 | 0.030 |
| Land use | 5 | 5.063 | 0.016 | 1.12 | 0.010 |
| Lithology | 4 | 4.010 | 0.003 | 0.90 | 0.004 |
| Elevation | 7 | 7.248 | 0.041 | 1.32 | 0.030 |
| Distance from river | 8 | 8.292 | 0.042 | 1.41 | 0.030 |
| Plan curvature | 3 | 3.009 | 0.005 | 0.58 | 0.010 |
| CI | 5 | 5.087 | 0.022 | 1.12 | 0.020 |
| HSG | 3 | 3.203 | 0.102 | 0.58 | 0.018 |

To derive the flood potential index across the study area, the AHP weights, together with the raster dataset associated with the flood predictors, were used in map algebra of ArcGIS software. The normalized values of FPI were classified into five classes using the natural break method. The very low class, between 0 and 0.12, covered about 18.9% of the total area and was mainly spread along the valleys located in the southern part of the catchment. Another 23% of the delimited zone was characterized by a low flood potential. The medium FPI values (between 0.32 and 0.53) were associated with about 33.3% of the delimited perimeter. The high and very high potential was spread around 24.8% and was associated with FPI values higher than 0.53 (Figure 8b).

## 5. Discussion

In the last ten years, the study of the susceptibility to hydrological risk phenomena has developed significantly as a result of the combined application of geospatial analysis techniques with statistical models or algorithms from artificial intelligence [49]. It is well known that small river basins located in mountainous areas favor the occurrence of flash-floods and their propagation toward the areas located in the lower zones of the basins. The mountainous area of Romania is not an exception and is often affected by severe flash-flood events that generate property damage and loss of life. In this context, the present study aimed to identify the areas exposed to floods generated by flash-floods within the Izvorul Dorului River basin located in the Romanian Carpathians, which could produce pollution such as the transport of polycyclic aromatic hydrocarbons resulting from different sources [57].

The present study included a first part in which the potential for rapid water runoff on the slopes was determined, the second part referred to the identification of valleys with high torrential potential, followed by the evaluation of flood susceptibility existing along these valleys. The potential for rapid surface runoff, expressed through the FFPI, was calculated by applying three ensemble models resulting from the combination of three statistical bivariate methods and the fuzzy AHP model.

The decision to apply three ensemble models was taken after a careful review of the literature according to which hybrid models have higher performance than stand-alone ones [15]. The models applied for the calculation of the FFPI showed that the hydrographic basin of the Izvorul Dorului River has a high and very high potential for a rapid surface runoff with a percentage between 38% and 58% of its surface. It was also highlighted that in particular, the upper and middle basin is characterized by these values of FFPI. Since the genesis of rapid water runoff on the slopes is finally reflected in the flooded areas along the rivers, it was decided to continue the study with the identification of valleys with a high

potential for flash-flood propagation, along with the identification of the floodplains. In this regard, the three FFPI models were evaluated, and the result provided by FAHP-WOE, characterized by an AUC-ROC curve of 0.837 in the case of training data and 0.79 in the case of test data, was identified as the most accurate. Using the methodology proposed by Costache et al. [25], the valleys in the study area were identified and classified according to the degree of torrentiality. Valleys with a small and very small propagation potential were eliminated from the analysis, with only those characterized by a medium, high, and very high potential being used. The AHP model was further used to calculate the flood potential index along the torrential valleys and at the same time to determine the potential for flooding generated by the flash-flood propagation. It is worth mentioning that following the flash-flood genesis (which is facilitated by the torrential areas highlighted in Figure 1) and their propagation, the areas located along the torrential valleys are the most affected regions because the water flow from the slopes will be concentrated on the main river network. Therefore, it is very important to indicate the surfaces that are finally affected by these complex phenomena. This resulted in 24% of the delimited surface having a high and very high potential for flooding.

In a previous study, Costache et al. [58] estimated only the flooding susceptibility along the large river valleys within the Trotuș River basin from Romania, unlike the present study which analyzed the following three elements in close spatial and causal connection: (i) flash-flood potential at the slopes level; (ii) river valleys torrential degree; and (iii) flood potential along the river basins within this small catchment. In addition to this difference regarding the methodological approaches, the present study also differed from that conducted by Costache et al. [58] by the methods proposed for determining the susceptibility to the analyzed hydrological hazards. Thus, in the present study, three ensemble models of the fuzzy analytical hierarchy process with bivariate statistical methods for the estimation of flash-flood potential at the slopes level were applied, while in the previous study, three other ensemble models of the adaptive neuro-fuzzy inference system (ANIFS) were applied to determine the flooding potential at the large river valley level. Moreover, the flow accumulation procedure was applied in the present research in order to identify the torrential valleys. Another example where the fuzzy multicriteria decision making analysis was proposed to estimate the flood susceptibility was the study carried out by Azareh et al. [59]. In that research, which focused on the Haraz watershed in Iran, the authors used a combination between DEMATEL, analytical network process, and fuzzy logic in order to estimate the flood susceptibility. Like in the present case, the performance of the applied model was very good, which was revealed by an AUC-ROC curve between 0.8 and 0.9. Nevertheless, the main difference between the present study and the research work developed by Azareh et al. [59] is given by the fact that the mentioned study only included the evaluation of the terrain surface potential along the river valley, to produce the flooding phenomenon and did not also include an evaluation of the slopes for surface runoff genesis.

## 6. Conclusions

The assessment of flash-floods and flood susceptibility is an actual scientific topic due to the high potential of the studies to propose solutions for reducing the economic damage and diminishing the number of victims. The new approach developed in the present research is useful because it provides a complete overview regarding the susceptibility of the entire phenomenon composed of rapid surface runoff on the slopes, the propagation of flash-floods generated by the surface runoff, and the potential for flooding along torrential valleys. The water quality in the floodplains will be lower because the flash-flood waves will be accompanied by the massive transport of materials from the slopes and inside the forest vegetation. Furthermore, the decision-makers will have a clearer image regarding the places they must implement measures to reduce the water runoff on the slopes, to arrange the torrential valleys, and to protect the areas exposed to floods.

**Author Contributions:** Conceptualization, R.C., Q.B.P. and A.B.; Data curation, R.C. and Q.B.P.; Methodology, R.C. and A.B.; Writing—original draft R.C., Q.B.P. and A.B.; Writing—review and editing, R.C., Q.B.P. and A.B. All authors have read and agreed to the published version of the manuscript.

**Funding:** This work was supported by a grant of the Romanian Ministry of Education and Research, CNCS-UEFISCDI, project number PN-III-P1-1.1-PD-2019-0424, within PNCDI III.

**Institutional Review Board Statement:** Not applicable.

**Informed Consent Statement:** Not applicable

**Conflicts of Interest:** The authors declare no conflict of interest.

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
