# Peer review of "Integrated Framework for Detecting the Areas Prone to Flooding Generated by Flash-Floods in Small River Catchments"

_water, doi:10.3390/w13060758_

Round 1

Reviewer 1 Report

I carefully read the manuscript water-1032019 untitled "Integrated framework for detecting the areas prone to flooding generated by flash-floods in small river catchments". The author proposed an interesting strategy monitoring flash-floods area. This is an important research topic worth of further investigation.

I find the manuscript well structure and interesting to read. The abstract is explicative and keywords are appropriate. The research method and data processing/analysis technique follow well-established routines and complying with state-of-art methodologies. I felt the discussion section and figures need  further improvement for better clarification.

I recommend for publication after minor modification to the text and figures.

Author Response

Dear Reviewer, 

Reviewer 2 Report

December 10, 2020

Dear Editor, dear Authors:

This manuscript entitled “Integrated framework for detecting the areas prone to flooding generated by flash-floods in small river catchments” presents results from a flash-flood susceptibility analysis based on statistical-models ensembles and propagation analysis. Although the topic fits the scope of the journal and might meet the interest of the scientific community, there is little scientific novelty in the presented analysis. In addition, in its actual form, the manuscript is not ready for publication and needs to be improved before to be reconsidered for publication in Water MDPI. Comments are reported below:

  • Even if I am not an English-native speaker, I would recommend an English edit to improve sentence structure and terminology. The text is difficult to read at some points and several sentences need to be modified.
  • The introduction is not well structured and does not provide a comprehensive summary of the problem analyzed in the text (i.e. the need for an integrated framework to analyze flash-flood susceptibility and potential alternative methods). Information provided in the introduction are of dramatic importance since they form a basis to interpret results of the analysis.
  • Section 2 should be part of the method section or should be removed.
  • The description of the study area is very short and, in my opinion should provide additional data about past flash floods, also in terms of frequency, propagation through the analyzed basin and effects.
  • Subsection 3.2 should be moved in the method section.
  • I would suggest to simplify the text shortening the less relevant part. Potentially, section 2 might be removed since such kind of methods are becoming of common use and are substantially known.
  • I would suggest to include an interpretation of the result in the context of the implication in land planning support perspective. In its actual form, the discussion section provides an interpretation of the potential of the applied method without a consistent comparison with further studies available in the literature. This is needed to underline the effective contribution of the paper. A further element would be a comparison between modeled susceptibility and known past events also in terms of propagation.
  • Please add a separated and short conclusion section.

Best regards.

Author Response

Dear Reviewer, 

Reviewer 3 Report

This paper titled “Integrated framework for detecting the areas prone to flooding generated by flash-floods in small river catchments” presents the results of the application of a series of GIS-based bivariate statistics to assess flood susceptibility in a small river basin in Romania. Authors demonstrate solid knowledge of the presented geospatial statistically-based modelling work, replicating methods previously used in published works. Nevertheless, authors seem not have implemented in this new submission (that I already reviewed in the previous submission rounds) all the efforts needed to make it worth for a publication on Water that is a high IF journal for this sector and topic.

In particular, the introduction is still very poorly structured and developed. The language is not coincise and not scientifically sound. A novel methodology, the missing calibration and validation, and the fact that this method should not be generalized and replicable by other researchers are factors of relevant importance that are completely missing in this work. The manuscript, in essence, is characterized by significant flows that are supporting my decision to confirm the major revision.

The main general comments are inserted here below, while specific comments can be found in the attached commented PDF.

General comments

  • The introduction is missing to properly introduce the framework, novelty and methods of this research submission. Authors shall edit and improve the introduction. Flash floods shall be properly introduced considering the several components and issues characterizing this research topic. See also specific comments in the text
  • On the methodology: authors claims “It should be mentioned that it is for the first time in the literature when the susceptibility of these two phenomena, flash-floods and flooding generated by them, are analyzed in an integrated way and in a spatial causal relationship”. Only in the discussion they self-cite the work Costache et al. (2019) as a base for the presented methodology. In the introduction it was never mentioned this work and there is no link with all the relevant assumptions and claims made in the discussion as for example “This new approach is useful because it provides a complete overview regarding the susceptibility of the entire phenomenon composed of rapid surface runoff on the slopes, the propagation of flash-floods generated by the surface runoff, and the potential for flooding along torrential valleys.” Authors are just using combined geospatial methods based on multi-variate statistics of geomorphometric and hydrologic parameters for estimating flood indicators. They mention that those areas were affected by torrential rains and floods, but there is no calibration and validation of the modelling as respect to observations. The training and validation data that are mentioned are not properly explained later on.
  • On the generalization and replicability of methods and results. Following general remark n.2, the paper is not only not providing and calibration and validation (even if observations are mentioned and, thus, available to authors), but is also missing to provide relevant information and methods for its generalization and replicability. The combination of geospatial modelling used (basically a list and combination of arcgis functionalities is not structured in a script that other researchers may use). Replicating this method elsewhere would be impacted by several user-based decisions that would make this method a useless student exercise. For example authors use an input layer in the modelling (a crucial one) like the stream network distance deciding not to consider the tributaries (see figure 4.d). This is just an example.

Reviewer 4 Report

  1. check carefully literature since some names are wrongly written, for example: StĘpnik - no capital letter in the middle
  2. add 2-4 pictures of your the Izvorul Dorului River (or its tributaries) so one have the impression how the river looks like
  3. consider adding the referencem: 1. Bonakdari, H.; Moradi, F.; Ebtehaj, I.; Gharabaghi, B.; Sattar, A.A.; Azimi, A.H.; Radecki-Pawlik, A. A Non-Tuned Machine Learning Technique for Abutment Scour Depth in Clear Water Condition. Water 2020, 12, 301. Water 2020, 12(1), 301; https://doi.org/10.3390/w12010301     and    2. Sattar, A.M.A.; Bonakdari, H.; Gharabaghi, B.; Radecki-Pawlik, A. Hydraulic Modeling and Evaluation Equations for the Incipient Motion of Sandbags for Levee Breach Closure Operations. Water 2019, 11, 279. Water 2019, 11(2), 279; https://doi.org/10.3390/w11020279 and 3. B. Gharabaghi, A. A. Sattar Gene Expression Programming in Open Channel Hydraulics, Chapter in Open Channel Hydraulics, River Hydraulic Structures and Fluvial Geomorphology For Engineers, Geomorphologists and Physical Geographers Edited by A. Radecki-Pawlik, S. Pagliara, J. Hradecký, E. Hendrickson, 2017, CRC Press, https://doi.org/10.1201/9781315120584
  4. check English more carefully

Author Response

Response to Reviewer #4

Comment 1: “check carefully literature since some names are wrongly written, for example: StĘpnik - no capital letter in the middle”

Authors: We thank the reviewer for the comment. We double-check the names of the authors and we observed that the names are written correctly. Most probably there are some letters specific for the country of that authors.

Comment 2: “add 2-4 pictures of your the Izvorul Dorului River (or its tributaries) so one have the impression how the river looks like”

Authors: We thank the reviewer for this valuable suggestion. Unfortunately, it is very hard for the authors to travel in the area of the study in the present moment for some photos because in that region the surface is covered with a thick layer of snow and the risk of avalanches is very high. Also, being a continuous snow layer the river geometry are not visible. Nevertheless, we will take into account this valuable suggestion for our future scientific works.

Comment 3: “consider adding the referencem: 1. Bonakdari, H.; Moradi, F.; Ebtehaj, I.; Gharabaghi, B.; Sattar, A.A.; Azimi, A.H.; Radecki-Pawlik, A. A Non-Tuned Machine Learning Technique for Abutment Scour Depth in Clear Water Condition. Water 2020, 12, 301. Water 2020, 12(1), 301; https://doi.org/10.3390/w12010301     and    2. Sattar, A.M.A.; Bonakdari, H.; Gharabaghi, B.; Radecki-Pawlik, A. Hydraulic Modeling and Evaluation Equations for the Incipient Motion of Sandbags for Levee Breach Closure Operations. Water 2019, 11, 279. Water 2019, 11(2), 279; https://doi.org/10.3390/w11020279 and 3. B. Gharabaghi, A. A. Sattar Gene Expression Programming in Open Channel Hydraulics, Chapter in Open Channel Hydraulics, River Hydraulic Structures and Fluvial Geomorphology For Engineers, Geomorphologists and Physical Geographers Edited by A. Radecki-Pawlik, S. Pagliara, J. Hradecký, E. Hendrickson, 2017, CRC Press, https://doi.org/10.1201/9781315120584”

Authors: We thank the reviewer for the valuable suggestion. We added all the suggested references.

Comment 4: “check English more carefully”

Authors: We thank the reviewer for the comment. We double-check the english in our mansucript.

Round 2

Reviewer 2 Report

Dear Editor, dear Authors,

the paper has been improved from the first version and can be now considered for publication in Water, MDPI.

Best regards.

Author Response

Response to Reviewer #2

General Comment:

“Dear Editor, dear Authors,

the paper has been improved from the first version and can be now considered for publication in Water, MDPI.

Best regards.”

Authors: We thank the reviewer for accepting our paper for publication.

Reviewer 3 Report

A rebuttal letter is needed to understand how authors addressed the major remarks.

Remark n.3 not addressed.

Authors still say "It should be mentioned that it is for the first time in the literature when the susceptibility of these two phenomena, flash-floods and flooding generated by them, are analyzed in an integrated way and in a spatial causal relationship"

Authors keep disregarding without even motivating the remarks. This is not respectful and is against the peer review process

If authors keep disregarding the concerns and remarks of reviewers and MDPI Water will publish this paper I'll personally share on social network the review process of this work.

Author Response

Response to Reviewer #3

Comment 1: “A rebuttal letter is needed to understand how authors addressed the major remarks.”

Authors: We thank the reviewer for the taking his valuable time to revise our mansucript.. We added new revisions for our manuscript and also we added a rebuttal letter.

Comment 2: “Remark n.3 not addressed.”

Authors: We thank the reviewer for the valuable comment. We mentioned in our manuscript which are the main elements of novelty in the Introduction section. Please see the text below:

“It should be mentioned that it is for the first time in the literature when the susceptibility of these two phenomena, flash-floods and flooding generated by them, are analyzed in an integrated way and in a spatial causal relationship. The previous studies carried out in Romania, as well as in any part of the globe, are focused on the estimation of flooding or flash-floods susceptibility without taking into account their strong spatial relationship” -  line 81

Comment 3: “Authors still say "It should be mentioned that it is for the first time in the literature when the susceptibility of these two phenomena, flash-floods and flooding generated by them, are analyzed in an integrated way and in a spatial causal relationship"”

Authors: We thank the reviewer for the comment. We still believe that we should keep this sentence in our manuscript because from our knowledges it is for the first time in the literature when the susceptibility of these two phenomena, flash-floods and flooding generated by them, are analyzed in an integrated way and in a spatial causal relationship.

Comment 4: “Authors keep disregarding without even motivating the remarks. This is not respectful and is against the peer review process.”

Authors: We apologize for any inconvenience. We have motivated our point of view in our the previous revision rounds.

Comment 5: “If authors keep disregarding the concerns and remarks of reviewers and MDPI Water will publish this paper I'll personally share on social network the review process of this work.”

Authors: We hope to resolve this issue in an amicable manner.